# A Real-World-Oriented Multi-Task Allocation Approach Based on Multi-Agent Reinforcement Learning in Mobile Crowd Sensing

**Junying Han [1,2], Zhenyu Zhang [1,2,*] and Xiaohong Wu [1,2]**

[1]  College of Information Science and Engineering, Xinjiang University, Urumqi 830046, China; sakurahjy@163.com (J.H.); wxh@xju.edu.cn (X.W.)

[2]  Xinjiang Multilingual Information Technology Laboratory, Xinjiang University, Urumqi 830046, China

*  Correspondence: zhangzhenyu@xju.edu.cn

**Abstract:** Mobile crowd sensing is an innovative and promising paradigm in the construction and perception of smart cities. However, multi-task allocation in real-world scenarios is a huge challenge. There are many unexpected factors in the execution of mobile crowd sensing tasks, such as traffic jams or accidents, that make participants unable to reach the target area. In addition, participants may quit halfway due to equipment failure, network paralysis, dishonest behavior, etc. Previous task allocation approaches mainly ignored some of the heterogeneity of participants and tasks in the real-world scenarios. This paper proposes a real-world-oriented multi-task allocation approach based on multi-agent reinforcement learning. Firstly, under the premise of fully considering the heterogeneity of participants and tasks, the approach enables participants as agents to learn multiple solutions independently, based on modified soft Q-learning. Secondly, two cooperation mechanisms are proposed for obtaining the stable joint action, which can minimize the total sensing time while meeting the sensing quality constraint, which optimizes the sensing quality of mobile crowd sensing (MCS) tasks. Experiments verify that the approach can effectively reduce the impact of emergencies on the efficiency of large-scale MCS platform and outperform baselines based on a real-world dataset under different experiment settings.

**Keywords:** mobile crowd sensing; multi-task allocation; multi-agent reinforcement learning; real-world-oriented

---

## 1. Introduction

### 1.1. Molile Crowd Sensing

Mobile crowd sensing (MCS) is an innovative paradigm of sensing based on crowd sourcing. MCS leverages smart mobile devices (smart phone, wearable device, etc.) that ordinary people carry to form a large-scale perception system, which can complete large-scale sensing tasks that traditional static sensing devices cannot solve. It has stimulated a lot of attractive applications, such as air quality monitoring, traffic information mapping, and infrastructure inspection. With the acceleration of urbanization and the development of smart cities, MCS has become a frontier research issue in the field of pervasive computing [1]. Smart community is one kind of typical pervasive computing environment.

The term mobile crowd sensing (MCS) was coined by Gant et al. [2] in 2011, who introduced a new data collection method by leveraging mobile terminals such as smartphones. Compared with traditional data collection technologies, MCS has some unique characteristics. First, the mobile devices have more computing, communication, and storage capability than mote-class sensors. Second, by leveraging the mobility of the mobile terminal users, the deployment cost of specialized sensing

infrastructure for large-scale data collection applications would be largely reduced. Currently, MSC has been widely used in many applications, including environmental monitoring [3], transportation [4], social behavior analysis [5], healthcare [6], and others [7–9], which demonstrates that MCS is a useful solution for large-scale data collection applications. In general, the typical MCS system includes three types of objects: (1) data requester; (2) MCS platform; (3) participants. In addition, the general workflow of mobile crowd sensing is as follows:

1)	The data requester sends a request to the platform. The data request will be designed by the MCS platform as a corresponding mobile crowd sensing task;
2)	Based on the real-time information of users and tasks, the platform adopts a certain method to achieve task allocation;
3)	Participants move to the target area to perform MCS tasks and upload data to the platform;
4)	The platform receives and processes the data which are uploaded by participants. In addition, the platform pays participants a certain amount of rewards.

The typical MCS system framework considered in this paper is shown in Figure 1. There are a series of things needed to be done in MCS. The process always consists of four main steps: assigning sensing tasks to mobile terminals, executing the task on the mobile terminals, collecting, and processing sensed results from the crowd. Obviously, assigning sensing tasks to mobile terminals is the primary issue to deal with the following steps. Therefore, the main issue and work is the task allocation in the MCS. The analysis and related work of the task allocation in the MCS will be introduced in the next section.

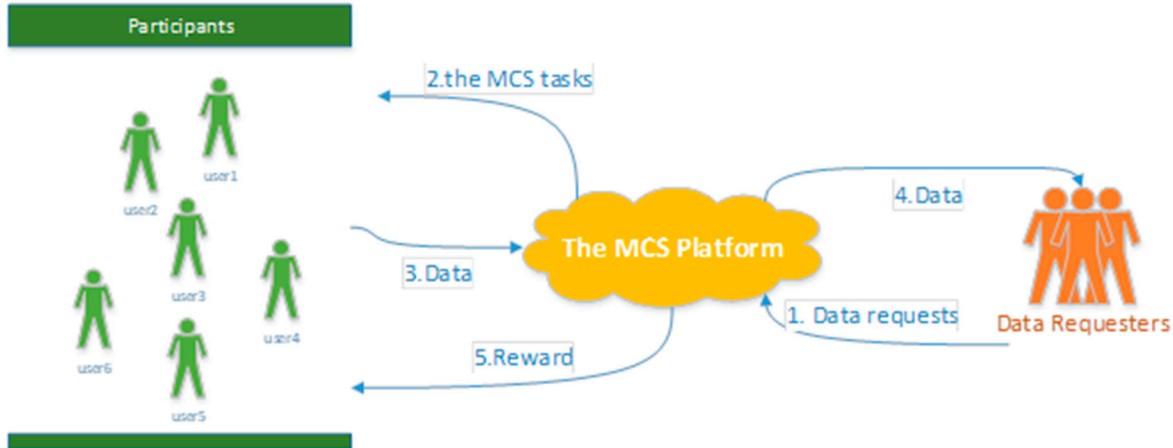

**Figure 1.** Typical mobile crowd sensing (MCS) system framework.

### 1.2. Task Allocation in the MCS

Task allocation is a key issue in mobile crowd sensing, which has an impact on the efficiency and quality of sensing tasks. Up until now, task allocation has been one of the major challenges in mobile crowd sensing. In addition, task allocation determines whether the perception based on mobile crowd sensing can be applied in real-world scenarios efficiently and flexibly.

Task allocation is a typical optimization problem which is inherently NP-hard and subject to external constraints, such as time, space, platform cost, and privacy protection. As a result, there are often multiple optimization goals in the task allocation under different situations. Designing an efficient optimization algorithm that can balance multiple optimization goals has always been the focus of existing research. Wang J et al. [10] redefined the multi-task allocation problem, and assigned each worker a set of appropriate tasks based on task-specific minimum perceptual quality thresholds. Liu et al. [11] designed the participant selection framework TaskMe for multi-task scenarios. Xiong et al. [12] selected participants with the aim to maximize the coverage quality of the

sensing task while satisfying the incentive budget constraint, but their work is not about multi-task allocation. Reddy et al. [13] studied a recruitment framework to identify appropriate participants for data collection based on geographic and temporal availability. Zhang et al. [14] proposed a novel participant selection framework, aiming to minimize incentive payments by selecting a small number of users with satisfying probabilistic coverage constraint. Cardone et al. [15] studied how to select participants to maximize the spatial coverage of crowd sensing.

However, a lot of previous task allocation approaches mainly focus on selecting a proper subset of users for a single task and lack the mechanism to deal with emergencies, which cannot be applied to the real-world scenarios effectively.

The real-world scenarios of MCS are complex, and the characteristics are as follow:

1) Multi-task real-time concurrency. In a typical MCS architecture, there is a centralized platform to publish MCS tasks and collect sensing data. Data requests come from different objects, such as government, individual user, scientific research department. Therefore, in a certain period of time, an MCS platform will encounter multi-task real-time concurrency. (Considering the above situation, we design an optimal path for each participant to accomplish multiple tasks, which minimizes the total sensing time while optimizing the quality of MCS tasks);

2) The heterogeneity of participants. Above all, the participants vary in mobility by taking different vehicles. In addition, the participants vary in ability to complete MCS tasks due to the device power, device capacity, storage space, etc. In other words, the number of tasks that each participant can actually perform is different, and the sensing time cost is determined by the distance and the speed, which greatly increases the difficulty of multi-task allocation;

3) The heterogeneity of tasks. The data requesters are different, and include governments, enterprises, and individual users. Therefore, the tasks vary in priority and importance weight. For example, the real-time perception of accidents requested by the government is more important than the user's perceived task. On the other hand, MSC tasks vary in the number of required participants and the target location;

4) Poor participant resources. Since the target area of the MCS task is uncertain, there may not be enough participant resources in the area of low population density. In the above case, all tasks are not guaranteed to be performed. In addition, under constraints of cost, the performed tasks cannot be completely accomplished. How to optimize the efficiency and quality of the tasks in the case of poor participant resource is a very challenging problem;

5) Accident and emergency. There are uncertain factors in the execution of the MCS tasks. In particular, two typical accidents are studied: (1) participant cannot reach the designated task area according to the planned path due to traffic jam; (2) participant quits halfway due to equipment failure, network paralysis, dishonest behavior, etc. The above accidents have an adverse effect on the completion of tasks and reduce the overall efficiency of the MCS platform.

All in all, a real-world-oriented multi-task allocation approach is significant for large-scale application of MCS.

### 1.3. Reinforcement Learning

Reinforcement learning is a highly universal machine learning method which is widely used in various fields, such as game theory, simulation optimization, and crowd intelligence.

The goal of reinforcement learning is to obtain an optimal policy in which the agent selects action under state. In addition, Q-learning is a classic reinforcement learning algorithm, and it can estimate Q value via the Time Difference method (TD method). In general, Q value is the expectation of the cumulative reward brought about by performing a certain action in a certain state. How to estimate the Q value accurately and quickly is one of the core tasks in reinforcement learning. At the present, the previous reinforcement learning approaches mainly consider single agent and single optimal

solution, which means there is only one agent in the environment space, and the agent only learns a single solution for the target problem.

### 1.4. Contributions in This Work

The key contributions of this paper include three parts: (1) we investigate the impact of different factors on multi-task allocation in real-world scenarios, and mainly take the heterogeneity of the participants and tasks into consideration to formulate an optimization problem which aims to minimize the total sensing time under meeting the sensing quality constraint; (2) we turn the quality optimization problem into a multi-pack problem whose optimal solution will be set as the sensing quality constraint; (3) we propose a novel multi-agent reinforcement learning framework. Firstly, the framework trains the reinforcement learning model, which can make a single agent learn multiple policies based on soft Q-learning, independently for every participant. Secondly, the framework designs two multi-agent cooperation mechanisms based on social convention and the greedy algorithm, which can help multi-agents obtain joint action.

## 2. Problem Model

### 2.1. Problem Formalization

Given a crowd of active participants in the MCS platform, denoted by the set $U = \{u_1, u_2 \cdots u_n\}$. The participant has three attributes: initial location, moving speed, number of tasks that can be performed, which are denoted by triples $u_i = (ip_i, ms_i, ne_i)$. MCS tasks have four attributes: fixed times, target location, importance weight, and execution time, denoted by tuple $t_j = (ft_j, tl_j, iw_j, st_j)$, and the task set is $T = \{t_1, t_2 \cdots t_m\}$. $\lambda_i$ is the task sequence performed by $u_i$. The total time $TU_i$ includes the participant's traveling time and the execution time of the tasks which are accomplished by $u_i$.

For task allocation, the relationship between the number of performers and the number of required participants in the MCS is a significant factor that influences the completion and efficiency of the sensing tasks. If the number of participants is too large and exceeds the number of people required for the task itself, data redundancy will occur, which is not conducive to analysis and affects the final perception quality of the sensing tasks. Therefore, we want to allow each sensing task to be assigned to the suitable number of participants to avoid data redundancy. In addition, the important tasks should be completed as much as possible, while taking the difference of the importance and priority of tasks into consideration. Therefore, the quality function is proposed to evaluate the quality and completion of the sensing tasks. Then, the quality function can be formulated as Equation (1):

$$Val(t_j) = \begin{cases} iw_j \cdot \frac{et_j}{ft_j}, et_j \leq ft_j \\ iw, et_j > ft_j \end{cases}, \tag{1}$$

where $et_j$ is the number of performers for task $t_j$. In the real-world scenarios, the multi-task allocation has two optimization goals, that are formulated as Equations (2) and (3):

$$\min \sum_{i=1}^{|U|} TU_i, \tag{2}$$

$$\max \sum_{j=1}^{|T|} Val(t_j). \tag{3}$$

The two optimization goals are in conflict. Therefore, we transformed the original problem into a single-objective optimization problem based on hierarchical target optimization theory. We will optimize the other goal as much as possible while satisfying one. Generally speaking, quality is a factor that the mobile crowd sensing platform should prioritize, so we optimize the sensing time when

we satisfy the sensing quality constraint. The ultimate optimization goal is formulated as Equations (4) and (5):

$$Goal(\Lambda) \triangleq min \sum_{i=1}^{n} TU_i \Big| Constraint^{SQ}, \tag{4}$$

subject to:

$$Constraint^{SQ} \triangleq max \sum_{j=1}^{m} Val(t_j), \tag{5}$$

where $\Lambda = \{\lambda_1, \lambda_2 \cdots \lambda_n\}$ is the set of sensing task sequences which is performed by participants.

### 2.2. Problem Analysis

There are three main challenges in the multi-task allocation which is applied in the real-world scenarios: (1) How to enable a single agent to learn multiple solutions. It is necessary for a single agent to master multiple policies, so that they can have the ability to cooperate and adjust their action for responding to emergencies; (2) How to get $Constraint^{SQ}$ to achieve the goal that optimizes the sensing quality, which is a complex problem; (3) Multi-agent cooperation. Up to now, avoiding the uncertainty and conflict is a key issue in multi-agent reinforcement learning. How to obtain a good enough joint action is a challenge.

## 3. Proposed Approaches

In order to generate sensing task sequences $\Lambda$ for multi-task allocation, a multi-agent reinforcement learning framework is proposed. The framework includes two parts: (1) Parallel training single agent. The framework trains the reinforcement learning model, which can make a single agent learn multiple solutions based on modified soft Q-learning independently for each participant; (2) Multi-agent cooperation. The framework designs two multi-agent cooperation mechanisms based on social convention and the greedy algorithm, which can help multi-agents to obtain joint action while meeting $Constraint^{SQ}$.

The multi-agent reinforcement learning framework is shown in Figure 2.

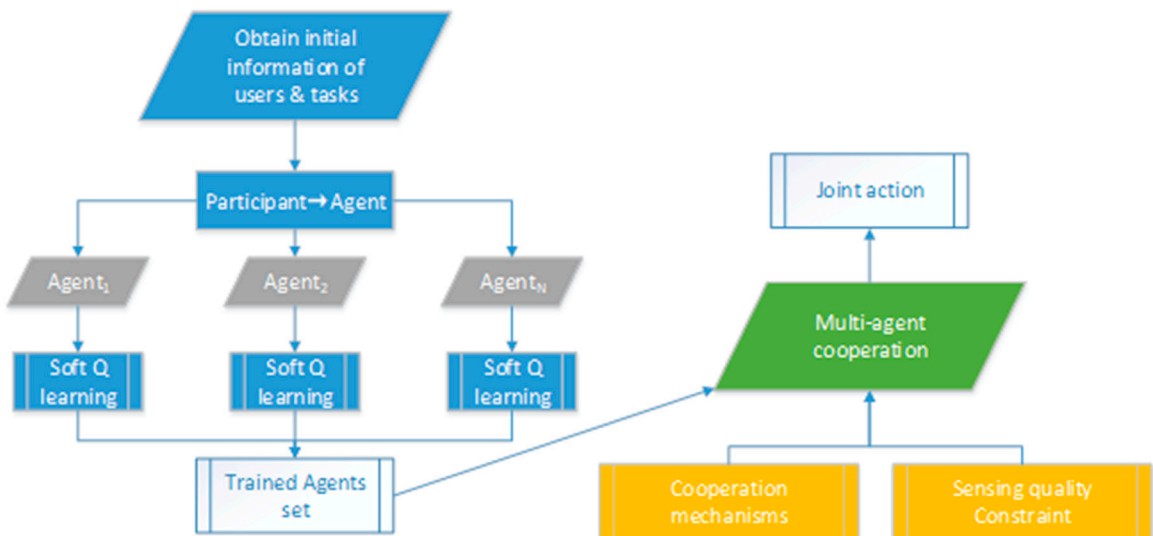

**Figure 2.** Multi-agent reinforcement learning framework.

### 3.1. An RL Model for a Single Agent.

Different from general reinforcement learning models, we hoped that our model could learn multiple solutions instead of a single optimal solution, so that the agents can adjust their behavior to

adapt to changes in the environment. Multi-agent cooperation, sensing quality constraints, emergencies, and other factors will limit the agent's action space and cause the agent to fail to perform optimal solutions in many scenarios. Therefore, agents must learn multiple solutions so that they can achieve cooperation or guarantee the perceived quality.

The reinforcement learning problem can be defined as policy search in an infinite-horizon Markov decision process, which can be represented by a four-tuple that includes the state space, the action space of the agent, the state transition probability, and the reward space. Through the interaction between the agent and the environment, reinforcement learning conducts trial and error learning, which learns the policy of the agent finally. In the case of multi-task allocation, the action space of agent is the task set, and the state is represented by the participants' current locations and the completion of the tasks. The time spent by participants is rewarded as a negative value. We can define the standard reinforcement learning objective as Equation (6):

$$\pi_{std}^* = \underset{\pi}{argmax} \sum_t [r(s_t, a_t)], \tag{6}$$

where $\pi_{std}^*$ is the optimal strategy to get the maximum cumulative reward, $s_t$ is the state at time $t$, and $a_t$ is the action made by the agent at time $t$. Table 1 shows a simple agent demo. There are five MCS tasks in this demo. Therefore, the action set of the agent is the unexecuted task set. The state consists of two parts: the current location of the user and the situation of task completion. For example, the tuple (0,0,0,0,0,0) represents that the user is in the initial position and none of the five tasks have been executed. In addition, the tuple (1,1,0,0,0,0) represents that the user is in the target area of the first task and the first task has already been performed.

**Table 1.** A simple agent demo.

| Time | State | Actions | Selected Action | New State |
|------|-------|---------|-----------------|-----------|
| 1 | (0,0,0,0,0,0) | (1,2,3,4,5) | 1 | (1,1,0,0,0,0) |
| 2 | (1,1,0,0,0,0) | (2,3,4,5) | 3 | (3,1,0,1,0,0) |
| 3 | (3,1,0,1,0,0) | (2,4,5) | 5 | (5,1,0,1,0,1) |
| 4 | (5,1,0,1,0,1) | (2,4) | 2 | (2,1,1,1,0,1) |
| 5 | (2,1,1,1,0,1) | (4) | 4 | (4,1,1,1,1,1) |

The standard reinforcement learning can only find an optimal policy to get the maximum reward and the agents cannot learn multiple solutions. To solve this, based on maximum entropy reinforcement learning, we augmented the reward with an entropy term, such that the optimal policy aims to maximize its entropy at each visited state, which ensures that agents can conduct sufficient exploration in the new state. We can define the maximum entropy reinforcement learning objective as Equation (7):

$$\pi_{maxEnt}^* = \underset{\pi}{argmax} \sum_t [r(s_t, a_t) + \alpha H(\pi(\cdot|s_t))], \tag{7}$$

where $H$ is entropy, and $\alpha$ is an optional but convenient parameter that can be used to determine the relative importance of entropy and reward. If the $\alpha$ is 0, the policy of the maximum entropy reinforcement learning is equal to the standard equation.

Equation (7) is a feasible way to train stochastic policies. In addition, we represented distribution of $\pi(a_t|s_t)$ with energy-based form. The distribution of the policies can be formulated as Equation (8):

$$\pi(a_t|s_t) = P(a_t|s_t) \propto exp(-\varepsilon(s_t, a_t)), \tag{8}$$

where $\varepsilon(s_t, a_t)$ is an energy function which determines the probability that agents execute a certain action in the current state. Generally speaking, value functions and Q-functions determine $\pi(a_t|s_t)$.

What is more, in the existing studies, soft Q learning is an effective method to achieve maximum entropy reinforcement learning. The standard soft Q learning is used in continuous action spaces, but the whole action space in the case of multi-task allocation is discrete. Therefore, we modified the standard soft Q learning to make it suitable for multi-task allocation in mobile crowd sensing. The modified optimal policy can be defined as Equations (9)–(11):

$$\pi^*_{maxEnt}(a_t|s_t) = exp\left(\frac{1}{\alpha}\left(Q^*_{soft}(s_t,a_t) - V^*_{soft}(s_t)\right)\right), \tag{9}$$

subject to:

$$V^*_{soft}(s_t) = \alpha \log \sum_a e^{1/\alpha \cdot Q^*_{soft}(s_t,a)}, \tag{10}$$

$$Q^*_{soft}(s_t,a_t) = r_t + \gamma V^*_{soft}(s_{t+1}). \tag{11}$$

$Q^*_{soft}(s_t,a_t)$ represents the expectation of the sum of the cumulative reward and entropy. For convenience, we will call it the Q value. In our reinforcement learning model, every agent has a stored table, named Q Table, which is generated to record the Q value. The agent will execute a certain action according to the Q value in the Q Table. If during the training process there is not enough exploration of the action space, then the Q value in the table will be too sparse, so that the agent can only learn a single solution instead of multiple solutions. Therefore, modified soft Q-learning based on maximum entropy reinforcement learning, which ensures that the agent has enough exploration during the training process, can help the agent learn multiple solutions for multi-agent cooperation and emergency response. We summarize an overview of modified Soft Q-learning in Algorithm 1.

---
**Algorithm 1:** Modified Soft Q-learning

---
**Input:** the user set $U$, the task set $T$
**Output:** the Q Table set Q Tables
Initialize the Q Tables is $\varnothing$
**for** episode e = 1 **to** E **do**
    $s_t \leftarrow s_0$
    Initialize the Q Table let the Q value is 0
    **while** the number of performed tasks is not equal to the number of executable tasks **do**
        $V^*_{soft}(s_t) = \alpha \log \sum_a e^{1/\alpha \cdot Q^*_{soft}(s_t,a)}$
        $\pi^*_{maxEnt}(a_t|s_t) = exp\left(\frac{1}{\alpha}\left(Q^*_{soft}(s_t,a_t) - V^*_{soft}(s_t)\right)\right)$
        $Q^*_{soft}(s_t,a_t) = r_t + \gamma V^*_{soft}(s_{t+1})$
        Update $s_t \leftarrow s_{t+1}$
**end for**
Q Tables join Q Table
parallel execute the above code for each user in $U$

---

### 3.2. Obtain the Sensing Quality Constraint

We should obtain the sensing quality constraint to control multi-agent cooperation for maximum sensing quality. The sensing quality constraint named *Constraint*$^{SQ}$ can be defined as Equation (12):

$$Constraint^{SQ} = \{qt_1, qt_2 \cdots qt_m\} = \underset{\{qt_1,qt_2\cdots qt_m\}}{argmax} \sum_j^m Val(t_j), \tag{12}$$

where $qt_j$ is the number of times task $t_j$ was executed. It is difficult to search the optimal solution directly. Therefore, we transformed it into a multi-pack problem, which is defined as a given set of items with value, volume, quantity, and a pack with a fixed maximum weight, and to gain the

maximum total value of the items placed in the pack without exceeding the maximum load of the pack. The multi-pack problem can be formulated as Equations (13)–(16):

$$Multipack(V, W, N, TW) \triangleq Constraint^{SQ}, \tag{13}$$

subject to:

$$TW = \sum_{i=1}^{|U|} ne_i, \tag{14}$$

$$V = \left\{ v \middle| v = \frac{iw_t}{ft_t}, t \in T \right\}, \tag{15}$$

$$N = \left\{ n \middle| n = min(MaxNE, ft_t), t \in T \right\}, \tag{16}$$

where *MaxNE* is the maximum number of times the task was executed, *W* is the value set of each item, which is initialized by a weight of 1, *N* is the set of number of items, *TW* is the maximum weight of the pack. Since there are already many methods to solve the pack problem, the method of solving the pack problem is not the focus of this article, so it will not be described in detail. In this article, the problem is solved by dynamic programming for obtaining the optimal solution (*Constraint^{SQ}*). We summarize an overview of dynamic programming for multipack problem in Algorithm 2.

---

**Algorithm 2:** Dynamic programming for multipack problem

---

**Input:** the user set *U*, the task set *T*
**Output:** *Constraint^{SQ}*
Initialize *MaxNE, W, N, TW* according to the *U&T*
Initialize *F*
**for** $i = 1$ **to** m **do**
    **for** $v = TW$ to 0 **do step** $-1$
        $F[i, v] = max\left(F[i-1, v - k * W[i]] + V[i] \middle| 0 \le k \le N[i]\right)$
    **end for**
**end for**
Obtain *Constraint^{SQ}* according to *F*

---

### 3.3. Multi-Agent Cooperation for Optimal Joint Action

*Constraint^{SQ}* controls multi-agent cooperation for maximum sensing quality. In addition, we should design suitable cooperation mechanisms to eliminate conflicts of action between multiple agents for better multi-agent cooperation. Two cooperation mechanisms are proposed based on social convention and greedy policy.

#### 3.3.1. Cooperation Mechanism Based on Social Convention

If all agents are allowed to act at the same time, due to the actions selected by the agents possibly conflicting, in the absence of information communication, the task sequences based on the local information to minimize sensing time are often not globally optimal. What is more, parallel decision-making will lead to instability of the joint action.

The social convention stipulates the order in which all agents make decisions in a certain state. When an agent as participant chooses to execute a certain task, the information will be shared with other participants. Let the left participants make reference to other participants' relevant information when making decisions, so as to gather stable joint action.

In real-world scenarios, participants are heterogeneous, especially the moving speed. Therefore, the order in which participants perform action specified in the social convention needs to have rationality. Considering that in real-world scenarios, the distances between participants and tasks are not too disparate, the participants' speed has a more important impact on minimizing the total sensing time.

Therefore, the cooperation mechanism based on social convention sorts participants into ascending order according to speed, so that participants with low speed will preferentially perform nearby tasks. We summarize an overview of the cooperation mechanism based on social convention in Algorithm 3.

---

**Algorithm 3:** Cooperation mechanism based on social convention

---

**Input:** the user set $U$, the task set $T$, $Constraint^{SQ}$, Q Tables
**Output:** $\Lambda$
Sort $U$ in ascending order according to speed
$s_t \leftarrow s_0$
**while** participants can continue to execute tasks **do**
　　**for** $u$ **in** $U$ **do**
　　　　**if** $u$ can no longer continue the task **then**
　　　　　　$U \leftarrow U - u$
　　　　　　Add the $\lambda_u$ into the $\Lambda$
　　　　**end if**
　　　　$u$ select the task $t$ with the largest Q value in $Q\ Table(s_t, u)$ when meet $Constraint^{SQ}$
　　　　Update $s_t \leftarrow s_{t+1}$
　　　　Update $Constraint^{SQ}$
　　　　Add task $t$ into the $\lambda_u$
　　**end for**
**return** $\Lambda$

---

### 3.3.2. Cooperation Mechanism Based on Joint Action-Value Function

To quantify the value of joint action, we propose a joint action-value function $Q_{joint}(\tau, s)$, which is formulated as Equation (17):

$$Q_{joint}(\tau, s) = \sum_{i=1}^{n} Q(\tau_i, s), \qquad (17)$$

where $\tau_i$ is the action performed by $u_i$, and $\tau$ is the joint action. It is obvious that the joint action-value function is a sum of individual value functions, which has the advantage of this representation that a decentralized policy arises simply from each agent performing greedy action selection with respect to its Q value, while meet the sensing quality constraint. The optimal joint action can be formulated as Equation (18):

$$\tau^* = \underset{\tau}{argmax}\, Q(\tau, s). \qquad (18)$$

Obtaining the optimal joint action is an optimization problem. In order to solve it, a recursive algorithm is used to obtain all possible joint actions which meet the sensing quality constraint in a given state. The set of possible joint actions is $\Upsilon_s = \left\{ \tau^{p1}, \tau^{p2}, \cdots \right\}^s$. Then, we find top k actions according to the Q value for each individual action be selected in the joint action, where the k is equal to the number of times that the task needs to be performed. We summarize

an overview of the cooperation mechanism based on joint action-value function in Algorithm 4.

---

**Algorithm 4:** Cooperation mechanism based on joint action-value function

---

**Input:** the user set $U$, the task set $T$, $Constraint^{SQ}$, Q Tables
**Output:** $\Lambda$
$s_t \leftarrow s_0$
**while** participants can continue to execute tasks **do**
  Obtain the $\Upsilon_{s_t}$ depends on the recursion on $Constraint^{SQ}$
  **for** $\tau^p$ in $\Upsilon_{s_t}$ **do**
    **If** $u$ can no longer continue the task **then**
      $U \leftarrow U - u$
      Add the $\lambda_u$ into the $\Lambda$
    **end if**
    **for** $\tau_i^p$ in $\tau^p$ **do**
      $K \leftarrow \tau_i^p$
      Select the *Top K* actions of a certain users according to Q value
      Add actions (tasks) into the $\lambda_u$
    **end for**
  Update $s_t \leftarrow s_{t+1}$
  Update $Constraint^{SQ}$
  **end for**
**return** $\Lambda$

---

### 3.4. Emergency Response Mechanism

As can be seen from Figure 3, the task allocation system of MCS platform consists of three modules: the task allocation module, quality control module, and emergency response mechanism. During the initialization phase, the MCS platform collects information of users and tasks. Secondly, based on the information of tasks, the quality control module gains the $Constraint^{SQ}$ via Algorithm 3. At the same time, the platform starts training for single agent. Thirdly, the platform gets the initial task allocation plan via multi-agent cooperation. Then, the participants move to the target area and perform the MCS tasks according to the initial plan. When users perform tasks, the emergency response module monitors the entire process and judges if an unexpected event has occurred.

If accidents such as traffic jams or road damage occur, the participant will send a message to the MCS platform, and the response module will update the information of tasks and users. The quality control module recalculates $Constraint^{SQ}$ based on the real-time information of the rest of participants and the unfinished task. In addition, when solving the multipack problem, items can be decomposed and monotonically queued by binary coding so that optimization is performed to reduce the time complexity. The agents do not need to be retrained to ensure the timeliness of the emergency response mechanism. Then, the task allocation module will adjust the task allocation plan via multi-agent cooperation again. Finally, participants change their action according to the adjusted task allocation plan.

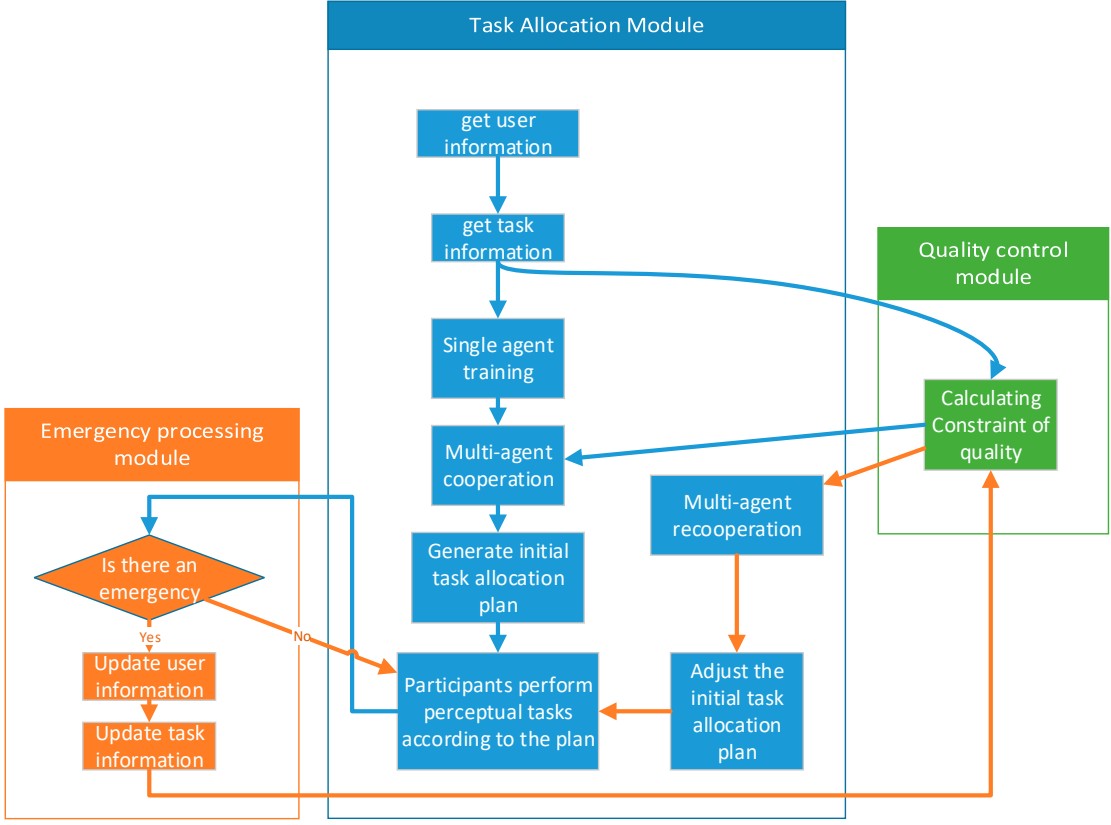

**Figure 3.** Task allocation system framework.

## 4. Experiment and Analysis

### 4.1. Experiment Data Preparation

The mobility dataset was provided by CRAWDAD, which was the GPS trajectory dataset of 500 taxis in San Francisco in the United States in 2008, which included the vehicle, latitude, longitude, occupancy (0 means taxi is not used, 1 means in use), time (Unix timestamp). Over the UNIX timestamp, the taxi location distribution in a certain period of time was taken as the set of MCS tasks that occurred concurrently. In addition, the unused state of the taxi was used as the participant set. The MCS tasks were heterogeneous in the complex real-world scenarios. Therefore, the importance weights of each task were randomly generated, and the sum of weights was 1. Participants were also heterogeneous. In terms of the speed of movement, considering the use of traffic vehicles, the speed of the user was randomly selected from 1, 3, 20, 60, which represents the moving speed of walking, bicycle, motorcycle, and car. The number of executions per user was also limited and heterogeneous. The number of executions was randomly generated between 3 and 20.

### 4.2. Evaluation of RL Model for Single Agentx

This paper proposed three baseline methods to compare with our RL model based on soft Q-learning. The baseline methods used the traditional reinforcement learning method Q-learning, Sarsa, and DQN (Deep Q Network), respectively, in the agent training. The experimental results of sparse degree are shown in Table 2. The sparse is the proportion whose Q value is approximately equal to zero.

As can be seen from Table 2, the Q values of the RL model based on traditional reinforcement learning approaches are sparse. Therefore, the RL model based on modified Soft Q-learning was relatively more efficient at ensuring that agents learned multiple solutions in the same environment based on Q value.

To verify the performance of the RL model based on Soft Q-learning more intuitively, we experimented with the same user and modified their speed. To simplify the figure content, we evaluated only SQL and QL. The results are shown in Figure 4.

**Table 2.** The sparse degree of Q value.

| Number of Tasks | Soft Q-Learnig | Q-Learning | Deep Q Network | Sarsa |
|:---:|:---:|:---:|:---:|:---:|
| 5 | 0.061 | 0.483 | 0.131 | 0.171 |
| 10 | 0.132 | 0.561 | 0.233 | 0.522 |
| 20 | 0.188 | 0.902 | 0.335 | 0.734 |

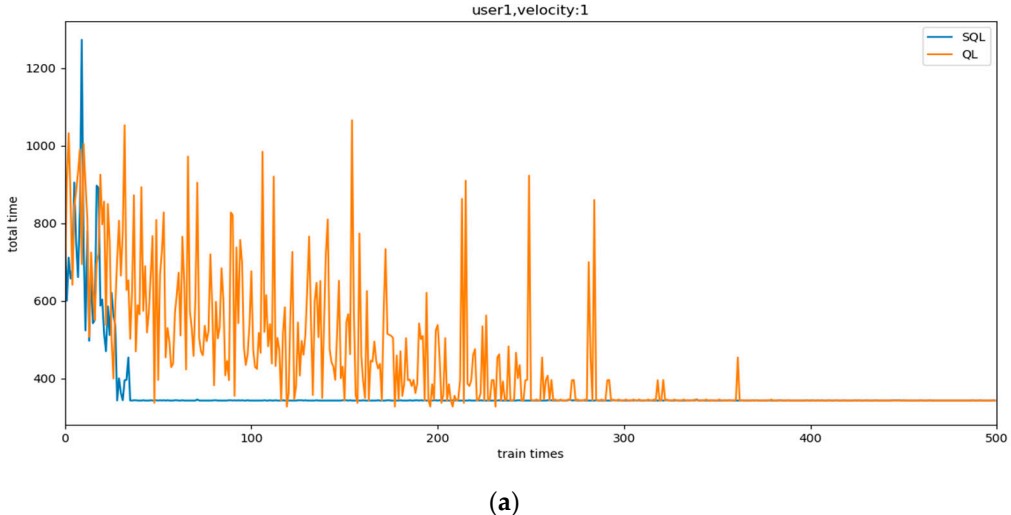

(**a**)

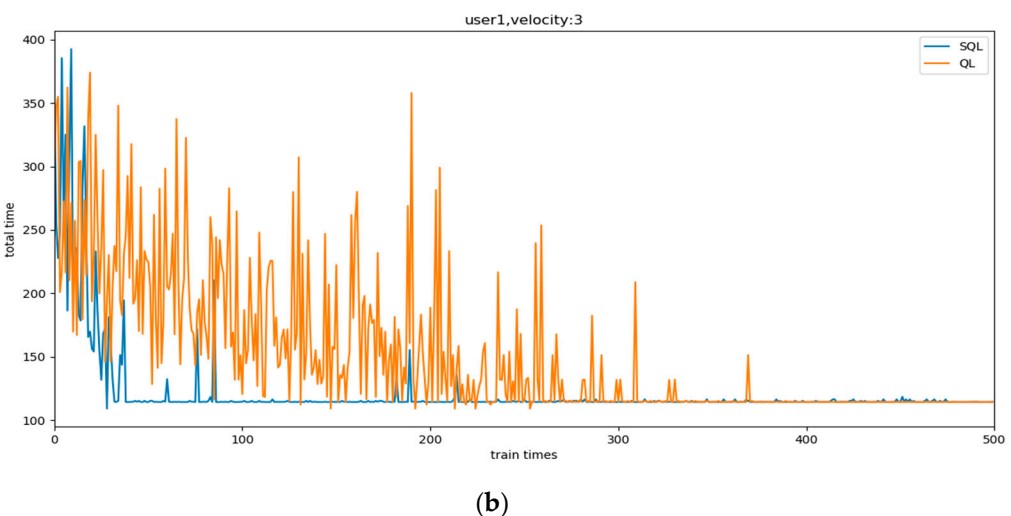

(**b**)

**Figure 4.** *Cont.*

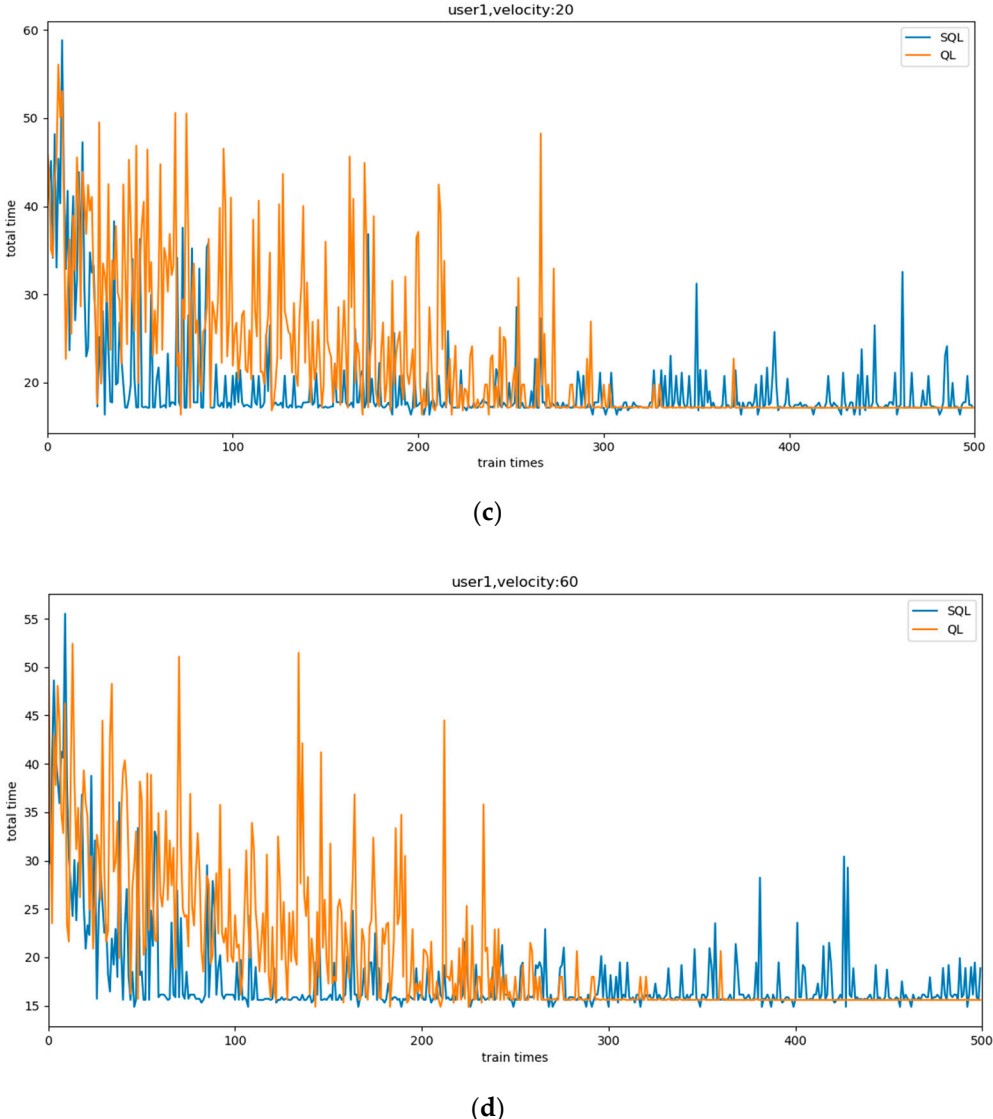

**Figure 4.** The comparison curve of total sensing time based on heterogeneous user: (**a**) the velocity: 1; (**b**) the velocity: 3; (**c**) the velocity: 20; (**d**) the velocity: 60.

We set four different speeds to represent user heterogeneity in the experiment. Obviously, SQL converged significantly faster than QL. In addition, we can see that the QL behaved consistently at different speeds, and the same action was always performed after convergence. On the other hand, SQL behaved differently at different speeds. As the speed increased, after the convergence, SQL still tried other actions regularly. This is because with the increasing speed the difference in the total reward brought by each action decreases, and SQL will execute actions with probability according to the Q value. While QL is hard to adapt to user speed changes due to sparse Q values and greedy policies, SQL can change policies based on user heterogeneity, making it possible for agents to learn multiple solutions.

### 4.3. Evaluation of Multi-Agent Reinforcement Learning Framework

In this section, we evaluate the performance of the overall multi-agent reinforcement learning framework in different experimental environments. The proposed baseline methods are shown in Table 3.

<div align="center">**Table 3.** Experimental baselines.</div>

| Name | Single RL Model | Cooperation Mechanism | The Sensing Quality Constraint |
|---|---|---|---|
| MARL-SSC | SQL | Social Convention | Multipack |
| MARL-SJQ | SQL | $Q_{joint}(\tau, s)$ | Multipack |
| MARL-DSC | DQN | Social Convention | Multipack |
| MARL-DJQ | DQN | $Q_{joint}(\tau, s)$ | Multipack |
| MARL-SR | SQL | Random queue | Multipack |
| MARL-SSCG | SQL | Social Convention | Greedy* |

Greedy* is a greedy heuristic algorithm to obtain the sensing quality constraint. In Greedy*, the participant greedily chooses the current maximum value according to the number of executable tasks. When the number of executable tasks reaches the upper limit, greedy selection is made in the remaining tasks until the task allocation is completed.

In the global scope, we set 100 sensing tasks, and the number of participants required for each task was a random number between 10 and 50. The task distributions were divided into three types based on the task location distribution: compact, scattered, and hybrid. Compact sensing tasks were more densely distributed and have relatively small relative distances. The scattered sensing tasks were more dispersed and relatively distant. The hybrid presented a compact distribution for some areas, and the remaining areas exhibited a scatter type distribution.

*Constraint*$^{SQ}$ determined the sensing quality directly. From Figure 5, the sensing of the quality of approaches with multipack was better. Therefore, it is an effective method to obtain the sensing quality constraint by transferring the origin problem into a multipack problem.

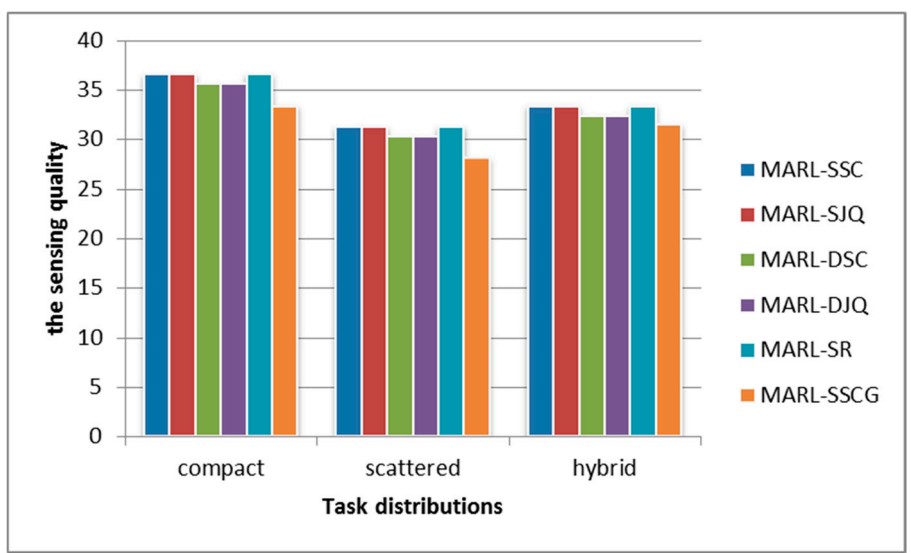

<div align="center">**Figure 5.** The sensing quality under different mobile sensing tasks distributions.</div>

As can be seen from Figure 6, the sensing time optimization effects of MARL-SSC, MARL-SJQ, and MARL-SSCG, whose single RL model was based on SQL, were significantly better than the other baseline methods. This is because in order to optimize the sensing quality as much as possible, all methods need to perform task allocation for satisfying the sensing quality constraint, which limits the action space of all agents. However, based on the SQL model and effective cooperation mechanisms, our multi-agent reinforcement learning framework could obtain better and more stable joint action under the above situation.

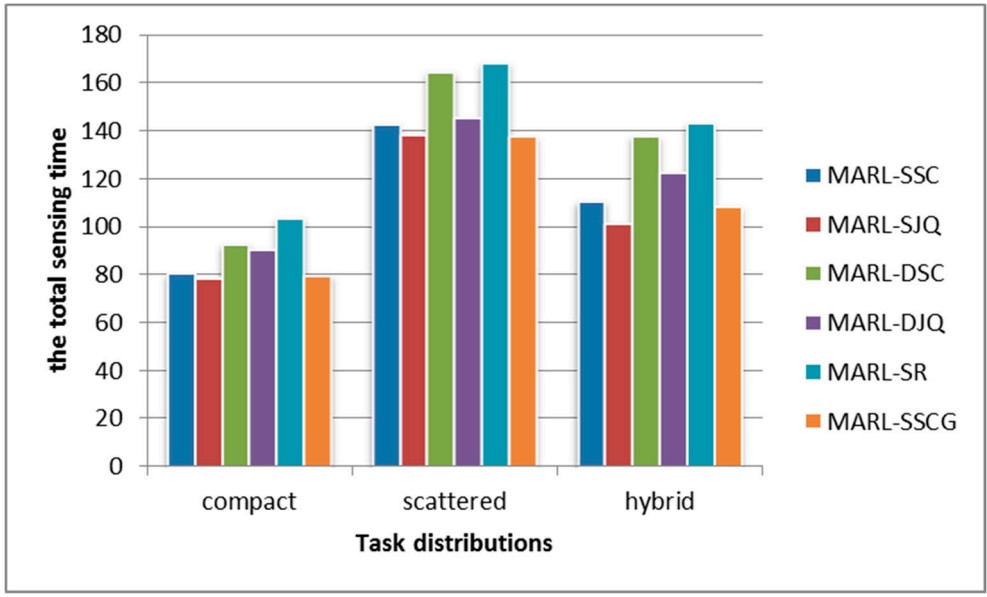

**Figure 6.** The total sensing time under different mobile sensing tasks distributions.

### 4.4. Evaluation of the MARL Framework in Case of Emergencies

A good enough real-world-oriented multi-task allocation requires the ability to quickly adjust and deal with emergencies. Therefore, we evaluated the performances of the baselines in the case of emergencies. The results are shown in Figure 7.

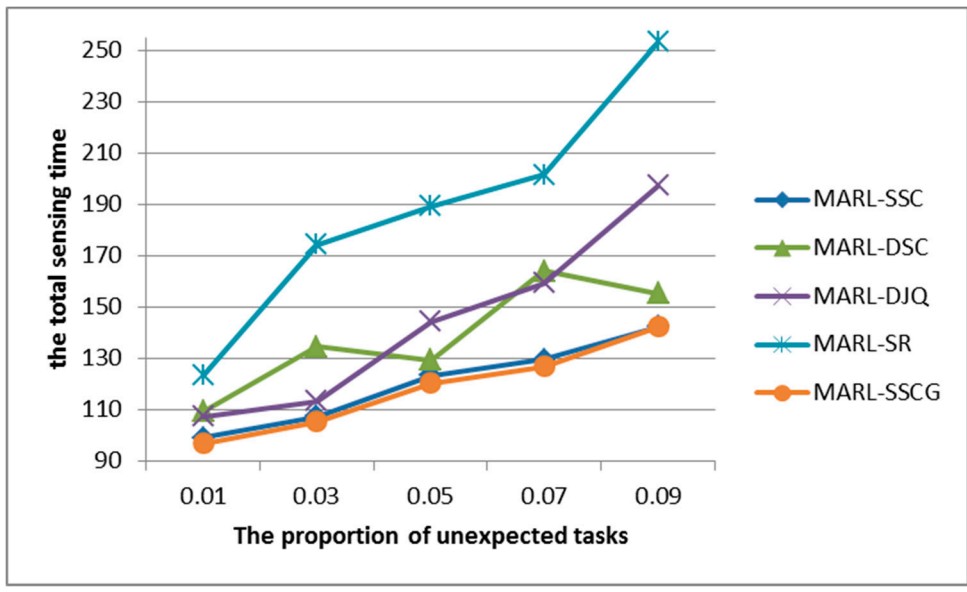

**Figure 7.** The total sensing time under different proportion of unexpected tasks.

The abscissa in Figure 7 is the proportion of unexpected tasks. The unexpected tasks refer to tasks that have been assigned in the initial scenario, but cannot actually be performed due to emergencies such as traffic jams. As can be seen from Figure 7, as the proportion of unexpected events increased, the multi-agent reinforcement learning frameworks based on SQL were less affected than other methods, and the growth range of sensing time was relatively small. Although the time of MARL-SR was always the largest, its growth curve was relatively stable without large fluctuations.

The MARL framework based on DQN fluctuated significantly, which is because unexpected events limited the agent's action space. When the agent is unable to greedily select the optimal solution due to

the sparse Q value, it is also unable to quickly adjust the policy to select the suboptimal solution. In a word, our MARL framework based on SQL has the ability to quickly adjust and deal with emergencies.

## 5. Conclusions

This paper mainly studied real-world-oriented multi-task allocation, and the key contributions include three parts: (1) we investigated the impact of different factors on multi-task allocation in real-world scenarios, and mainly took the heterogeneity of the participants and tasks into consideration to formulate an optimization problem which aims to minimize the total sensing time under meeting the sensing quality constraint; (2) we turned the quality optimization problem into a multi-pack problem whose optimal solution will be set as the sensing quality constraint; (3) we proposed a novel multi-agent reinforcement learning framework. Firstly, the framework trains the reinforcement learning model, which can make a single agent learn multiple policies based on soft Q-learning independently for every participant. Secondly, the framework designs two multi-agent cooperation mechanisms based on social convention and the greedy algorithm, which can help multi-agents to obtain joint action.

The proposed real-world-oriented multi-task allocation approach fully considers factors such as multi-task concurrency, task heterogeneity, user heterogeneity, and enables the sensing platform to respond to emergencies in the sensing process in time. In the future, we will further study multi-agent reinforcement learning, and strive to propose a multi-task allocation based on the willingness of users that is easier to implement and can be applied to large-scale real-world scenarios.

**Author Contributions:** J.H. and Z.Z. conceived the idea of the paper. J.H. and X.W. designed and performed the experiments; J.H. and Z.Z. analyzed the data; J.H. wrote and revised the paper. All authors have read and agreed to the published version of the manuscript.

**Funding:** This research was funded by the National Natural Science Foundation of China, grant number 61262089.

**Conflicts of Interest:** The authors declare no conflict of interest.

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
