# Peer review of "A Real-World-Oriented Multi-Task Allocation Approach Based on Multi-Agent Reinforcement Learning in Mobile Crowd Sensing"

_information, doi:10.3390/info11020101_

Round 1

Reviewer 1 Report

Summary:

The article "A Real-world-oriented Multi-Task Allocation Approach based on Multi-Agent Reinforcement Learning in Mobile Crowd Sensing" discusses task allocation approaches for mobile crowdsensing (MCS) scenarios in smart cities. First, related work and problems in the context of task allocation in MCS scenarios, including the occurrence of disruptions such as emergencies, as well as solutions utilizing reinforcement learning are discussed. Following this, a formal problem definition is defined. The authors then propose a task allocation approach based on multi-agent reinforcement learning. More specifically, modified soft Q-learning is used in order to enable agents of heterogeneous participants to learn multiple solutions for heterogeneous tasks. Furthermore, the authors propose two cooperation mechanisms to obtain the optimal joint action in terms of sensing quality while minimizing the total sensing time. Finally, experiments that evaluate the proposed approaches compared to baseline methods with a real-world dataset are described and discussed.

Points in favor:

The article covers an interesting and relevant topic The overall methodology is proper. All problem statements and algorithms are explicitly described and formalized. Most formalisms appear to be sound. The approach is evaluated by using (partly) real-world data.

Points against:

1: The authors introduce MCS very briefly and never define which type of MCS scenarios they target with their approaches. Smart cities and participatory sensing tasks seem to be the main focus of the research, but this should be explicitly stated.

Additionally, almost no related work in the context of MCS is introduced. It should be stated where the article is to be placed inside the broader topic, e.g.:

Ganti, Raghu K., Fan Ye, and Hui Lei. "Mobile crowdsensing: current state and future challenges." IEEE Communications Magazine11 (2011): 32-39.

Burke, Jeffrey A., et al. "Participatory sensing." (2006).

Guo, Bin, et al. "From participatory sensing to mobile crowd sensing." 2014 IEEE International Conference on Pervasive Computing and Communication Workshops (PERCOM WORKSHOPS). IEEE, 2014.

3: Acronyms and terms like DQN and Q value should be introduced the first time they are used.

Language and grammar are rather poor throughout the article and should be revised. For example:

Line 53: “proposed that the multi-task allocation framework”

Line 65: “which likes Amazon Mechanical Turk”

Line 91: “in order to make the MCS paradigm can be applied”

Line 99: “can only learn single solution”

Line 144: The sub-clause “Since the two optimization goals are in conflict” stands alone

Line 200: “which likes Boltzmann distribution”

Line 205: “enough good” Line 298-299: “according to the recalculated the sensing constraint”

Line 354: “we can find it that”

Line 361: “make them cannot execute”

Some of the equations are incorrect or are using variables that are never introduced,

e.g.: 1: the conditions are overlapping

6: s and a are not introduced Formalisms are used to an extent that makes it difficult to follow the text. The equations should be simplified and explained more clearly.

It remains unclear how the architecture of the described MCS platform is designed. At what point in time are agents trained? How does the platform communicate with its agents? This would be particularly important for the emergency response mechanism described in Section 3.3.

It remains unclear how and to which extent the real-world dataset is actually used for the evaluation, since importance weights, movement speed and number of executions are randomly generated.

Author Response

Thank you for your careful review and comments. Thank you for your hard work.

Reviewer 2 Report

The proposed approach is interesting because the authors try to apply it to real-world scenarios. The authors study how different factors (multi-task concurrency, task heterogeneity and user heterogeneity) affect the multi-agent reinforcement learning. The main factor taken into account has been the heterogeneity of agents and tasks. Sufficient experimental results are presented in order to support the conclusions: the proposed approach reduces the impact of emergencies on the efficiency of large-scale Mobile Crowd Sensing platforms. The problems is well contextualized and the results are exposed in an appropriate way. The conclusion are clear. The study is a good piece of work.

Author Response

(The authors gave the same response as above.)

Round 2

Reviewer 1 Report

Summary:

The authors have addressed many concerns that I have raised for the first version of the paper.

One concern should be still addressed. The paper should be improved in the writing with respect two to reasons:

Some statements are very difficult
e.g.,
(1) Mobile crowd sensing (MCS) is an innovative paradigm of sensing based on crowd sourcing; crowd soursing and crowd sensing are related, but not based on each other
(2) The first sentence of the abstract does not very well fit
(3) The sentence starting with Secondly in the abstract is hard to read
... Several minor language issues should be fixed, sometimes figure, sometimes Figure; Line 32: MCS leverage -> MCS leverages; Abstract: experiment settings -> experimental settings

In addition, in Section 1.2, more works should be cited, especially for the mentioned points in the bullet point list. To some more statements, also references should be added, e.g., "Smart community is one kind of typical
38 pervasive computing environment."

Author Response

Response to Reviewer 1 Comments (round2)

Thank you for your careful review and comments. Thank you for your hard work.

Point 1: Mobile crowd sensing (MCS) is an innovative paradigm of sensing based on crowd sourcing; crowd sourcing and crowd sensing are related, but not based on each other.

Response 1: We have modified the original content to make it easier to read and more reasonable. The modified content is as follows:’ Task allocation is a key issue in mobile crowd sensing. Although there have been some research results so far, multi-task allocation in the real-world scenarios is still a huge challenge.’

Point 2: The first sentence of the abstract does not very well fit

Response 2: We deleted the original content and replaced it with the following: Task allocation is a key issue in mobile crowd sensing.

Point 3: The sentence starting with Secondly in the abstract is hard to read.

Response 3: We deleted the original content and replaced it with the following: Secondly, the initial task allocation plan is obtained through the cooperation of multiple trained agents. Thirdly, an emergency response mechanism is proposed to adjust the initial task allocation plan in time when something unexpected happens.

Point 4: Several minor language issues should be fixed, sometimes figure, sometimes Figure; Line 32: MCS leverage -> MCS leverages; Abstract: experiment settings -> experimental settings.

Response 4: Thank you for your careful work, we have fixed these errors one by one.

Point 5: In addition, in Section 1.2, more works should be cited, especially for the mentioned points in the bullet point list.

Response 5: Based on your comments, we have added more references.
